# Single Core Genome Sequencing for Detection of both *Borrelia burgdorferi* Sensu Lato and Relapsing Fever Borrelia Species

**DOI:** 10.3390/ijerph16101779

**Published:** 2019-05-20

**Authors:** Sin Hang Lee, John Eoin Healy, John S Lambert

**Affiliations:** 1Milford Molecular Diagnostics, Milford, CT 06460, USA; 2School of Biological, Earth and Environmental Sciences, University College Cork, T23 N73K Cork, Ireland; e.healy@cs.ucc.ie; 3Department of medicine, University College Dublin, D04 V1W8 Dublin, Ireland; jlambert@mater.ie; 4Mater Misericordiae University Hospital, D07 R2WY Dublin, Ireland

**Keywords:** Lyme disease, core genome, DNA sequencing, borreliosis, relapsing fever borreliae, metagenomic diagnosis, *Borrelia*, genus-specific PCR primers, second PCR, direct detection, Sanger sequencing

## Abstract

Lyme disease, initially described as Lyme arthritis, was reported before nucleic-acid based detection technologies were available. The most widely used diagnostic tests for Lyme disease are based on the serologic detection of antibodies produced against antigens derived from a single strain of *Borrelia burgdorferi*. The poor diagnostic accuracy of serological tests early in the infection process has been noted most recently in the 2018 Report to Congress issued by the U.S. Department of Health and Human Services Tick-Borne Disease Working Group. Clinical Lyme disease may be caused by a diversity of borreliae, including those classified as relapsing fever species, in the United States and in Europe. It is widely accepted that antibiotic treatment of Lyme disease is most successful during this critical early stage of infection. While genomic sequencing is recognized as an irrefutable direct detection method for laboratory diagnosis of Lyme borreliosis, development of a molecular diagnostic tool for all clinical forms of borreliosis is challenging because a “core genome” shared by all pathogenic borreliae has not yet been identified. After a diligent search of the GenBank database, we identified two highly conserved segments of DNA sequence among the borrelial 16S rRNA genes. We further developed a pair of *Borrelia* genus-specific PCR primers for amplification of a segment of borrelial 16S rRNA gene as a “core genome” to be used as the template for routine Sanger sequencing-based metagenomic direct detection test. This study presented examples of base-calling DNA sequencing electropherograms routinely generated in a clinical diagnostic laboratory on DNA extracts of human blood specimens and ticks collected from human skin bites and from the environment. Since some of the tick samples tested were collected in Ireland, borrelial species or strains not known to exist in the United States were also detected by analysis of this 16S rRNA “core genome”. We recommend that hospital laboratories located in Lyme disease endemic areas begin to use a “core genome” sequencing test to routinely diagnose spirochetemia caused by various species of borreliae for timely management of patients at the early stage of infection.

## 1. Introduction

Lyme disease and related borreliosis transmitted by the bite of borrelia-infected ticks is the most common vector-borne infectious disease in the United States and in many European countries [1,2,3,4,5]. It is generally believed that at the early stage of infection this disease can be treated successfully with a proper course of antibiotics, but within days to weeks certain species of *Borrelia* may disseminate from the site of the tick bite to other regions of the body if not properly treated [1]. Misdiagnosed or untreated Lyme disease and related borreliosis may result in debilitating health consequences [6]. Historically, clinical Lyme disease in humans was thought to be caused exclusively by *Borrelia burgdorferi* sensu stricto in North America [7] and laboratory diagnosis was based on serology tests to detect antibodies in convalescent serum samples of Lyme disease patients against the antigens derived from the *B. burgdorferi* sensu stricto strain B31 [8,9]. In the United States, the consumers paid $492 million in 2008 to seven of the largest commercial laboratories for Lyme disease serology tests, according to an estimate made by the Centers for Disease Control and Prevention (CDC) [10]. However, it is now generally recognized that infections by many species of the relapsing fever borrelia group, for examples, by *Borrelia miyamotoi* [11,12], *Borrelia hermsii*, and *Borrelia turicatae* [13] may also cause clinical manifestations which are difficult to distinguish from those caused by *B. burgdorferi* sensu stricto. At least one sample in the serum repository panel classified by the CDC as reference serums from proven Lyme disease patients was found to contain *B. miyamotoi* instead of *B. burgdorferi*, by 16S ribosomal RNA (rRNA) gene sequencing [14]. The clinical manifestations of *Borrelia lonestari* infection may closely resemble those of “Lyme disease” [15]. Furthermore, infections by different members of the *B. burgdorferi* sensu lato complex may induce production of antibodies that do not fully match the epitopes of the antigens used in various serology test kits, thus leading to possible diagnostic failures [16,17,18]. Clinically, the reliability of using *Erythema migrans*, often referred to as “bull’s eye” skin rash, as the diagnostic feature of early Lyme disease is also in question due to its uncertain prevalence and specificity [19]. Hence, there is a need for a reliable test to detect all of the borrelial pathogens that can cause clinical “Lyme disease” while avoiding false-positive test results which may lead to inappropriate treatment with potentially serious complications [20,21].

A recent *Viewpoints* publication authored by stakeholders of regulatory agencies and healthcare industries with interests in Lyme disease diagnostics in the United States has acknowledged “Reliable direct-detection methods for active *B. burgdorferi* infection have been lacking in the past but are needed and appear achievable” [22]. While recognizing genomic sequencing as an achievable method for diagnosis, the *Viewpoints* article also stated “Challenges remain, however, including our incomplete knowledge of the full breadth of *Borrelia* genomic diversity, that is, of the genes that might be shared by all isolates (also called the “core genome”) and those that might be unique to specific species (also called the “accessory genome”). Without this critical knowledge, it would be challenging to determine precisely which genes or antigens should be targeted by a selective diagnostic test”. The authors of the *Viewpoints* article advocate achieving the goal of direct detection tests by using or planning to use “High-throughput sequencing (HTS), also known as next-generation sequencing (NGS)”. However, NGS is still an emerging, not yet stable technology for general use in disease diagnosis [23], and the first clinical trial to test the ability of NGS to detect *Borrelia burgdorferi* DNA in the blood of 20 study participants with an *Erythema migrans* rash only announced its patient recruitment from 28 July, 2018 [24]. Since the authors of the *Viewpoints* publication include several key officers of the CDC and the Food and Drug Administration (FDA), the viewpoints expressed in the article will have far-reaching effects on the future healthcare policies affecting Lyme disease patient management worldwide.

In bacteriology, the term “core genome” has been used to refer to a set of orthologous genes common to a species [25] or a genus [26,27]. In Lyme disease diagnostics, “core genome” is defined as the genes that might be shared by all Lyme borrelia isolates [22]. Testing for a core genome shared by all strains of one species of the *B. burgdorferi* sensu lato complex only will miss many other species of *B. burgdorferi* sensu lato and relapsing fever borreliae which are capable of causing “clinical Lyme disease”. For the lack of other options due to our incomplete knowledge of the full breadth of *Borrelia* genomic diversity [22], we propose using a highly conserved *Borrelia* genus-specific segment of the 16S rRNA gene as the diagnostic “core genome” for the detection of both *B. burgdorferi* sensu lato and relapsing fever borrelia species in clinical specimens and in vectors. Microbial 16S rRNA genes have been considered to be well-established cores [28] and conventional 16S rRNA gene phylogeny reconstruction has been used as a reference in bacterial core genome research [27].

The goal of this study was to present our evidence-based data to demonstrate that a highly conserved *Borrelia* genus-specific segment of the 16S rRNA gene can serve as a “core genome” shared by all species and isolates of *Borrelia*, including members of the *B. burgdorferi* sensu lato complex and the more heterogeneous species of the relapsing fever borreliae which are known to be capable of infecting the ticks and humans in North America and in Europe [1,2,3,4,5] as well as in Africa [29]. Even the louse-borne borrelia, *Borrelia recurrentis*, shares this highly conserved segment of 16S rRNA gene which can be detected, if present in the blood. The detected DNA sequence can be validated by comparing it with the standard borrelial DNA sequences catalogued in the GenBank in community hospital laboratories located in tick borne disease-endemic areas for accurate molecular diagnosis without the need of expensive software, personnel with bioinformatic expertise and computational resources which are required by all NGS platforms [30].

Methods based on DNA sequencing to circumvent the obstacles of obtaining pure culture for the detection of microorganisms, living or dead, are referred to as metagenomics [31]. Recently, the evolving concept of “one test diagnostic metagenomics” has begun to penetrate into the clinical microbiology laboratory [32]. Based on the latter concept, 16S rRNA gene DNA sequencing for detection and for genotyping of borreliae without prior culturing is within the scope of one test diagnostic metagenomics and depends on analysis of unambiguous base-calling DNA sequences generated by the diagnostic laboratories [33,34,35]. Since most clinical laboratories have not been performing in-house Sanger sequencing or do not report their diagnostic DNA sequences for borrelia identification, it is difficult to evaluate the publications retrieved from the literature on the subject of metagenomic DNA sequencing diagnosis of Lyme disease and related borreliosis. To date there have been no published studies on the specificity of any routine DNA-based direct detection method for the diagnosis of borreliae in human whole blood or sera, although there was one report of false-positive PCR testing for Lyme disease on human synovial fluid [21]. In the current study we chose to publish samples of the computer-generated Sanger sequencing electropherograms which were routinely included in the diagnostic reports as evidence of “good practice” in the era of precision medicine. The validity of the results of studying borrelia distribution in *Ixodes scapularis* ticks using PCR and DNA sequencing without supportive sequencing electropherograms [36] has been questioned because the DNA sequences claimed to have been detected were questionable [37]. In our study for the first time evidence was presented to show that Sanger sequencing of one PCR amplicon can detect not only pathogenic borreliae on North American samples but also for European borrelia species.

## 2. Materials and Methods

### 2.1. Search for PCR Primers for Borrelial Core Genome Amplification

The DNA sequences of 16S rRNA genes of borrelial species were retrieved from PubMed Nucleotide database [38]. The criteria for suitable core genome primer sites were two different segments of sequence, each of 21 nucleotides (bases) in length [39] defining a conserved segment of borrelial 16S rRNA gene DNA of about 300 bases long with variable regions for speciation of the most common pathogenic borreliae. To be useful for metagenomic diagnosis, the 21-base PCR primer sites must be substantially different in sequence from the corresponding sites of the 16S rRNA genes of common blood-borne pathogens [40]. After these potential primer sites were identified in the GenBank sequences, their corresponding forward and reverse primer pairs were synthesized by Sigma-Aldrich Life Science (USA) or Thermo Fisher Scientific (USA). The synthesized primers were first tested on DNA extracts of pure cultures of *Borrelia burgdorferi* sensu stricto (ATCC 53210) and *Borrelia coriaceae* (ATCC 43381), the two representatives of the *B. burgdorferi* senso lato complex and of the relapsing fever borrelia group, respectively, for their ability to amplify the intended segments of DNA derived from these two borrelial species, and to test if the PCR products could be used as the templates for Sanger sequencing for routine species differentiation as expected. Then the synthesized PCR primer pairs were further tested on blood specimens from patients visiting Milford Hospital seeking for diagnosis of Lyme disease with patients’ consents and hospital IRB approval at the initial method development stage before the test was approved for commercial diagnostics under Clinical Laboratory Improvement Amendments of 1988 (CLIA) and on ticks removed from human skin bites or collected from the environment for the detection of borrelia infections on these samples.

### 2.2. Sources of Borrelial Genomic DNAs

Crude DNA extracts from the following specimens were used for this study: (1)Frozen pure cultures of *Borrelia burgdorferi* sensu stricto strain B31 (ATCC 53210) and *Borrelia coriaceae* (ATCC 43381) in liquid media purchased from ATCC;(2)Venous blood specimens collected from U.S. patients submitted by licensed physicians for “Lyme bacteria” tests as part of routine patient management and 25 blind-coded simulated patient blood specimens from New York State (NYS) Department of Health (DOH) for a *Borrelia burgdorferi* proficiency test; the patients’ informed consent was not required because it is implied that informed consent was previously given for the scope of the treatment and because the patients’ identities were not revealed; ethical approval was not required because it is considered that this was not research but clinical/laboratory practice [41].(3)Blind-coded archived serum samples derived from patients with and without “Lyme disease” from the CDC under two Material Transfer Agreements (MTA No. NCEZID-R137154-00 and NCEZID-R147284-00);(4)Engorged *Ixodes scapularis* ticks removed from humans living in the United States and submitted for Lyme testing as part of preventive patient care; and(5)Unfed, questing *Ixodes ricinus* nymphs collected from free public access areas in Ireland by “flagging” in the counties of Kerry, Waterford, Galway, Wicklow, and Donegal (John Eoin Healy). Acquiring tick samples in areas of free public access in the Republic of Ireland for the purpose of laboratory analysis does not require approval from any state authorities or governmental agencies.

### 2.3. DNA Extractions

The cultures of *Borrelia burgdorferi* and *Borrelia coriaceae* purchased from ATCC were diluted to various densities in 0.85% NaCl and the spirochetes counted microscopically in a dark field chamber with a Petroff-Hausser grid. The dilution containing 1000 spirochetes per mL was used for crude DNA preparation to test the various synthesized PCR primer pairs. On the day of the experiment, a 100-µL aliquot of the diluted culture containing 100 spirochetes was mixed with 200 μL 0.7 M NH_4_OH (Sigma-Aldrich) in a 1.5-mL plastic microcentrifuge test tube. The mixture was heated at 95–98 °C for 5 min with closed cap, followed by 10 min with open cap. After the test tube was cooled to room temperature, 30 μL of 3 M sodium acetate and 700 μL of ice-cold 95% ethanol were added to the mixture. The mixture was centrifuged at ~16,000× *g* for 5 min and the supernatant discarded. The precipitate was re-suspended in 1 mL of ice-cold 70% ethanol. The suspension was centrifuged at ~16,000× *g* for 5 min. After all liquid was discarded, the pellet was air-dried, re-suspended in 100 µL TE buffer (Tris-EDTA buffer, Sigma Aldrich 93302) and heated at 95–98 °C for 5 min. The heated suspension was finally centrifuged at ~16,000× *g* for 5 min. The crude DNA extract in the supernatant was used as the template to initiate the PCR with various primer pairs.

The venous blood specimens collected from patients with “Lyme disease” in a lavender top test tube containing EDTA anticoagulant were shipped to the laboratory at ambient temperature via overnight courier delivery. The whole blood specimen was first centrifuged at ~400× *g* for 15 min to spin down the red and white cells. One mL of the platelet-rich plasma above the buffy coat was transferred to a 1.5-mL plastic tube to be further centrifuged at ~16,000× *g* for 10 min to collect the platelets and the spirochetes, if any, in the pellet. After the supernatant was discarded, the pellet was suspended in 100 µL of TE buffer, pH 7.4, and 200 μL 0.7 M NH_4_OH. The mixture was heated at 95–98 °C for 5 min with closed cap, followed by 10 min with open cap. The extracted crude DNA was precipitated in ethanol and dissolved in 100 µL TE buffer as described for the borrelial cultures. The NYS DOH proficiency whole blood samples were processed in the same manner as patient blood for DNA extraction.

The archived serum panels received from the CDC were 100 µL in volume for each blind-coded sample. The bacteria in the serum were pelleted by centrifugation at ~16,000× *g* for 10 min and the microbial DNA in the pellet was extracted in 100 μL TE buffer and 200 μL 0.7 M NH_4_OH and precipitated in ethanol, as described for the borrelial cultures. The DNA was finally dissolved in 100 μL TE buffer.

To test for borrelia DNA in ticks, each dried tick was placed in a 1.5-mL plastic tube and immersed in 300 μL of 0.7 M NH_4_OH overnight at room temperature. On the following day, the test tubes were heated at 95 °C to 98 °C for 20 min with closed caps, followed by 10 min with open caps. After the test tubes were cooled to room temperature and the carcass of the tick was discarded, 30 μL of 3 M sodium acetate and 700 μL of 95% ethanol were added to each NH_4_OH digestate. The precipitated crude DNA was spun down in the pellet after centrifugation at ~16,000× *g* for 5 min, washed in 1 mL of ice-cold 70% ethanol, air dried, and re-dissolved in 100 μL of TE buffer by heating the DNA extract at 95 °C to 98 °C for 5 min. The DNA in the crude extract was used for PCR amplification without further purification.

### 2.4. PCR Primers

One pair of M1 and M2 PCR primers was used to amplify a highly conserved 357/358 bp segment of the borrelial 16S rRNA genes for detection of both *Borrelia burgdorferi* sensu lato complex and the relapsing fever borrelia species. The PCR amplicon defined by the M1 and M2 primers was used as the template for routine direct Sanger sequencing. The sequences of these primers are: M1 (forward): 5′-ACGATGCACACTTGGTGTTAA-3′ andM2 (reverse): 5′-TCCGACTTATCACCGGCAGTC-3′

When further speciation was required, an adjacent 282 bp 16S rRNA gene DNA segment was amplified by a supplementary heminested PCR, and the heminested PCR products were used as the template for direct Sanger sequencing. The sequences of the heminested PCR primers were: Primary PCR Forward Bg1 primer: 5′- GACGTTAATTTATGAATAAGC-3′Primary PCR Reverse Bg6 primer: 5′- TTAACACCAAGTGTGCATCGT-3′Heminested PCR Forward Bg5 primer: 5′- CGGGATTATTGGGCGTAAAGGGTGAG-3′Heminested PCR Reverse Bg6 primer: 5′- TTAACACCAAGTGTGCATCGT-3′

### 2.5. PCR Conditions

All synthesized oligonucleotides which were initially considered to be useful candidate PCR primers for metagenomic detection of borrelial 16S rRNA genes were diluted to 10 μmolar working solutions in TE buffer. To initiate each PCR, 1 μL of 10 μmolar forward primer, 1 μL of 10 μmolar reverse primer, 20 μL of ready-to-use LoTemp^®^ PCR mix (HiFi DNA Tech, LLC, Trumbull, CT, USA), and 3 µL of the crude borrelial DNA extract of either *B. burgdorferi* or *B. coriaceae* pure culture containing about three copies of borrelial chromosomal DNA were mixed in a thin-walled PCR tube in a total 25 µL volume. To test the human blood, the CDC sera and the ticks, 3 µL of the crude DNA extracts from these samples as described above were used as the testing material without prior DNA purification and quantitation. The thermocycling steps were programmed to 30 cycles at 85 °C for 30 s, 50 °C for 30 s, and 65 °C for 1 min after an initial heating for 10 min at 85 °C, with a final extension at 65 °C for 10 min. The bands of PCR products were usually invisible at agarose gel electrophoresis. An internal nested (or heminested) PCR or a second PCR, previously referred to as same-nested PCR [14], was needed to re-amplify the first PCR products for a band of target DNA PCR products to be visualized at gel electrophoresis.

To perform a second PCR, the complete PCR mixture consisted of 20 μL of ready-to-use LoTemp^®^ PCR mix, 3 µL of water, and the same forward and reverse primers (1 μL each) as those used in the first (primary) PCR, in a total volume of 25 μL. A trace (about 0.2 μL) of the primary PCR products was transferred by a micro-glass rod (to avoid pipetting aerosol) to the complete second PCR mixture. The thermocycling steps were programmed to 30 cycles at 85 °C for 30 s, 50 °C for 30 s, and 65 °C for 1 min after an initial heating for 10 min at 85 °C, with a final extension at 65 °C for 10 min. An aliquot of 5µL was pipetted from all primary and nested PCR products for agarose gel electrophoresis to detect the bands of the target DNA amplicons if any.

When the DNA extracts were tested, a negative water control and a positive *B. coriaceae* DNA were always included for each PCR run. Sample preparation, DNA extraction, reagent preparation, nested PCR preparation, and Sanger sequencing were conducted in different rooms of the laboratory to minimize the chances of cross contamination in compliance with the standards set for CLIA-certified laboratories.

### 2.6. Sanger Sequencing

The positive nested PCR products without purification were transferred directly from the nested (second) PCR tube by a micro-glass rod into a Sanger reaction tube containing 1 μL of 10 μmolar sequencing primer, 1 μL of the BigDye^®^ Terminator (v 1.1/Sequencing Standard Kit), 3.5 μL 5× buffer, and 14.5 μL water in a total volume of 20 μL for 20 enzymatic primer extension/termination reaction cycles according to the protocol supplied by the manufacturer (Applied Biosystems, Foster City, CA, USA). After a dye-terminator cleanup with a Centri-Sep column (Princeton Separations, Adelphia, NJ, USA), the reaction mixture was loaded in an automated ABI 3130 four-capillary Genetic Analyzer for sequence analysis. Sequence alignments were performed against the standard sequences stored in the GenBank by on-line BLAST analysis [42]. To be useful for molecular diagnosis, the nested PCR primer pair must be able to amplify a conserved segment of the borrelial 16S rRNA gene of various *B. burgdorferi* sensu lato species and a corresponding conserved DNA segment of the heterogeneous relapsing fever borreliae as the template for Sanger sequencing. At least 100 unambiguous bases in sequence were required for each BLAST analysis. The readable DNA sequence must have a 100% ID match with a signature reference sequence of a borrelial 16S rRNA gene for a reliable molecular diagnosis of the causative agent.

## 3. Results

### 3.1. Genus-Specific PCR Primers for Metagenomic DNA Detection of Borrelia

After reviewing the alignments of the borrelial 16S rRNA gene sequences of the common species of *Borrelia* capable of causing human infections and after testing all the candidate primers synthesized, we identified a unique segment of *Borrelia burgdorferi* sensu lato 16S rRNA gene and the corresponding segment of 16S rRNA genes of various relapsing fever borreliae, including that of *B. recurrentis*, all defined by a pair of highly conserved 21-base DNA sequences. This segment of DNA is located at positions 736-1096 of the *Escherichia coli* (*E. coli)* 16S rRNA gene (Sequence ID: MK336731). The conserved 21-base sequences defining the 357/358 bases of the borrelial 16S rRNA gene segment are substantially different from those defining the corresponding DNA segments in *E. coli* and other bacteria analyzed by visual alignments. Among all the primer pairs synthesized and tested, this particular pair of primers was found to consistently generate a PCR amplicon of borrelial 16S rRNA genes from various sources for successful Sanger sequencing analysis, and was referred to as the *Borrelia* genus-specific or the borrelial core genome PCR primers for single PCR amplicon detection of borrelial DNA extracted from clinical specimens and from the ticks removed from patient’s skin bites or collected in the environment. The sequences of these two primers which have been used previously for the detection of *B. burgdorferi* and *B. miyamotoi* in human blood and serum samples [14,43] were referred to as M1 and M2 primers.

The highly conserved 357/358-base segments of the 16S rRNA gene sequences with variable regions of the 20 borrelial species commonly encountered in clinical practice are aligned in Figure 1, as follows.

In Figure 1, the sequences were presented in reverse complement direction with the M1 primer site at the end because primer M2 was used as a routine sequencing primer to show the most variable regions near the M1 primer site for speciation. Usually, the first 50–100 bases downstream of the sequencing primer in routine automated Sanger sequencing are difficult to decipher. When the M1 primer was used as the sequencing primer, the computer could not perform the base calling function in the entire electropherogram if the sample contained a *B. burgdorferi* sensu lato strain and a relapsing fever borrelia because there is a one-base gap due to frameshift indels in the 16S rRNA genes of the *B. burgdorferi* sensu lato species close to the M1 primer site, compared to those of the relapsing fever borreliae. The 357-base 16S rRNA gene sequences of some species of *B. burgdorferi* sensu lato, e.g., most strains of *Borrelia garinii*, and *Borrelia bavariensis*, as well as the corresponding 358-base sequences of the species of *Borrelia turicatae*, *B. hermsii*, and *Borrelia parkeri* are identical in this highly conserved segment.

As illustrated in Figure 1, shortening of the size of this PCR amplicon by moving the primers M1 and M2 further inwards may risk failure to detect some borrelial species. Moving the primers outwards to increase the size of the amplicon may risk reducing sensitivity of detection and may generate more non-specific PCR products. Examples of using the pair of M1/M2 PCR primers in the second PCR setting for metagenomic DNA sequencing detection of various borreliae in human blood samples and in ticks were illustrated as follows with the figures of all base-calling sequencing electropherograms placed in the Appendix A.

### 3.2. Routine Detection of B. burgdorferi Sensu Lato

Appendix A showed a bi-directional Sanger sequencing in routine metagenomic diagnosis of *B. burgdorferi* sensu lato in the United States.

### 3.3. Routine Detection of B. garinii Bernie Strain

Appendix A showed a bi-directional Sanger sequencing in routine metagenomic diagnosis of *B. garinii* Bernie strain infection in an *Ixodes ricinus* tick collected in Ireland.

### 3.4. Routine Detection of Borrelia valaisiana

Appendix A showed a typical Sanger sequencing segment of 16S rRNA gene of *B. valaisiana*.

### 3.5. Routine Detection of B. miyamotoi

Appendix A showed that Sanger sequencing of the M1/M2 PCR amplicon detected and differentiated two strains of *B. miyamotoi* based on single nucleotide polymorphism.

### 3.6. Routine Detection of Mixed Infection by B. burgdorferi and B. miyamotoi

Appendix A showed that visual analysis of the sequence of the M1/M2 PCR amplicon with M2 sequencing primer was capable of detecting mixed infection by both *B. burgdorferi* and *B. miyamotoi* species in human blood and in a tick.

### 3.7. Routine Detection of Other Relapsing Fever Borreliae

Appendix A showed an example of a relapsing fever borrelia (*B. coriacea*) 16S rRNA gene other than that of *B. miyamotoi*.

### 3.8. Supplementary Heminested PCR/Sequencing for Further Borrelial Speciation

As stated above, sequencing of the M1/M2 primer PCR amplicon cannot distinguish many *B. garinii* strains and *B. bavariensis* from *B. burgdorferi*, or distinguish among the species of *Borrelia turicatae*, *B. hermsii* and *B. parkeri*. An adjacent 282-bp heminested PCR amplicon was sequenced when there was a need for distinction among these borreliae.

The highly conserved 282-base segments of the 16S rRNA genes with single nucleotide polymorphisms of 19 borrelial species commonly encountered in clinical practice were aligned in Figure 2 as follows.

### 3.9. Bg5/Bg6 Primer Sequencing for Distinction between B. burgdorferi and B. garinii

The species of *B. garinii* is heterogeneous and most strains of *B. garinii* (except the Bernie strain) share a common 357-base segment of the 16S rRNA gene sequence with *B. burgdorferi* defined by the M1 and M2 PCR primers. Sequencing of the Bg5/Bg6 heminested PCR amplicon was capable of distinguishing *B. garinii* from *B. burgdorferi*, as demonstrated in Appendix A.

### 3.10. Potential Pitfalls in Metagenomic DNA Diagnosis of Lyme Borreliosis

The major obstacle in 16S rRNA gene sequencing diagnosis of *B. burgdorferi* spirochetemia is the low density of spirochetes in the circulating blood of the patients. Since there may be <20 spirochetes in 1 mL of plasma of the patients with Lyme disease infection [44] and there is only one copy of 16S rRNA gene per spirochete [45], a nested or a second PCR is generally required to amplify the target DNA brought into the test tube in order to generate a molecular mass of PCR products visible as a band at gel electrophoresis for detection and to be used as the template for Sanger sequencing. While nested PCR is a way to increase the sensitivity of target DNA detection, it is prone to DNA contamination from adjacent samples and from the positive control DNA which is always incorporated in every PCR run. However, cross contamination is not inherent in the nested PCR technology. It is rather a function of the laboratory operation. In our laboratory, the positive control is a strain of *B. coriaceae* which has a unique nucleotide in the borrelial 16S rRNA gene segment selected as positive control for pathogenic borrelia PCR detection (see Figure 1). Any positive control DNA contamination during the PCR process would be immediately recognized at the stage of Sanger sequencing by the reviewing pathologist.

In the absence of borrelial 16S rRNA genes as the preferred template, the M1 and M2 PCR primers may anneal to a segment of human genomic DNA which is partially matched with the sequence of the primers at the 3′ ends to initiate an unintended PCR. These unintended PCR products may form a visible band at nested PCR product electrophoresis and may cause confusions if the size of the unintended PCR amplicon is close to that of the target DNA. One of such examples is illustrated in Figure 3 and Appendix A.

On this gel image, the primary PCR products of samples 11-20 were practically invisible whereas each of the second PCR products of samples 14, 15, 17, 18, and 20 showed a distinct band of DNAs, migrating at a speed close to that of the positive control. Sanger sequencing showed that the nested PCR products in samples 15 and 20 were identified correctly to be positive for *B. burgdorferi*. All the other nested PCR products proved to be those of non-target DNA. N = negative, no sample control; P = positive *B. coriaceae* control. These results were reported to the NYS DOH and accepted as correct answers as the blind proficiency testing simulated patient blood samples #11–20 were coded by the NYS DOH as Table 1.

The five electropherograms of Sanger sequencing on the second (nested) PCR products illustrated in lanes 14, 15, 17, 18, and 20 (Figure 3) are presented in Appendix A for comparison. These sequencing results showed that the single band visualized in lane 14 (Figure 3) probably represented a mixture of multiple PCR amplicons of slightly different sizes with heterogeneous sequences that no one major single template was available for a successful Sanger reaction. Sanger sequencing showed that the PCR products illustrated in lanes 17 and 18 were those of human genomic DNA.

When fresh blood samples were tested and the differential centrifugation protocol was properly followed, there were rarely non-target PCR products visualized on gel electrophoresis. Since the blood samples used for this proficiency testing panel were not from fresh blood and were partially hemolyzed, some fragmented nucleated blood cells were spun down in the platelet pellet for NH_4_OH digestion, thus generating non-specific second PCR products in many borrelia-negative samples. Non-target DNA amplification is usually suppressed when borrelial genomic DNA is present in the PCR mixture [46].

### 3.11. Selection of the Borrelial 16S rRNA Gene Target Region for Routine Detection

All bacteria have at least one copy of 16S rRNA gene which consists of highly conserved nucleotide sequences interspersed with variable regions [40]. Theoretically, there are many options in the selection of PCR primers for the purpose of bacterial identification. For metagenomic DNA diagnosis of Lyme borreliosis, selecting the right segment of core genome shared by all species of potential pathogenic borreliae for PCR amplification is crucial for successful sequencing analysis [47,48]. As demonstrated in Appendix A, the Bg5 and Bg6 primer heminested PCR products often generated non-specific base-calling peaks at certain positions on the computer-generated sequencing electropherogram when relapsing fever borrelia 16S rRNA gene DNA was amplified, but not when *B. burgdorferi* sensu lato 16S rRNA gene DNA was amplified although the 16S rRNA gene DNA sequences between these two groups of borreliae are highly conserved, at least in this segment of genes selected for amplification. As a result, we chose the M1/M2 primer pair for routine direct detection amplification.

## 4. Discussion

### 4.1. The Need for a Reliable Test for all Pathogenic Borreliae in Blood Samples

Lyme disease was first reported in 1977 as “Lyme arthritis” of uncertain etiology [49]. At the time when Lyme disease was recognized as an infectious disease in 1982 [50], there were no nucleic acid-based tests for its diagnosis and the causative agents were difficult to culture in the artificial media available. The only laboratory test for Lyme disease was based on detection of the antibodies against the antigens derived from *B. burgdorferi* strain B31 during convalescence [8,9]. However, in view of the recent recognition that “clinical Lyme disease” may be caused by borreliae other than *B. burgdorferi*, even by relapsing fever borreliae, notably *B. miyamotoi*, the practice of relying on antibody tests for the diagnosis of “Lyme disease” or borreliosis is now open to question [22]. There is an urgent need for a reliable test to detect all pathogenic borreliae in the blood specimens for early diagnosis of borreliosis, not just for the infection by *B. burgdorferi* or by *B. miyamotoi*.

### 4.2. Sequencing of one PCR Amplicon for Detection of all Pathogenic Borreliae

In the current study, we have presented examples of using one single pair of *Borrelia* genus-specific PCR primers to amplify a segment of the 16S rRNA “core genome” for metagenomic detection of various pathogenic borreliae and to prepare the templates for Sanger sequencing. In addition to the borrelial species listed in Figure 1, other members of the *B. burgdorferi* sensu lato complex, e.g., *Borrelia bissettii* (NR_102956), *Borrelia americana* (HM802226), and *Borreliella californiensis* (NR_148824), also share this conserved segment of 16S rRNA gene, according to the data retrieved from the GenBank. The M1/M2 genus-specific primers also define a highly conserved segment of the 16S rRNA genes of other not listed relapsing fever borreliae, such as *Borrelia venezuelensis* (MG651649) and *Borrelia* sp. Qtaro (LC382043). These less common and unlisted borreliae would be detected if present in the specimens being tested. Since there is one nucleotide short in the M1/M2 primer-flanked segment of all *B. burgdorferi* sensu lato 16S rRNA genes, compared to those of the relapsing fever borreliae (Figure 1), routine Sanger sequencing can easily distinguish the detected DNA sequence of a *B. burgdorferi* sensu lato from that of any relapsing fever borreliae. Most of the common pathogenic borreliae can be diagnosed by analysis of a single PCR amplicon consisting of a 357/358-base 16S rRNA gene sequence defined by the M1 and M2 primers by comparing it against the reference sequences stored in the GenBank. Based on literature search, the authors have not found any 16S rRNA gene isolated from potentially pathogenic borrelia which does not bear the M1 and M2 primer sequences as shown in the DNA sequence alignments in Figure 1. However, these findings do not rule out the possibility that there may be pathogenic borrelial strains whose 16S rRNA genes may not have the fully matched M1/M2 primer binding sequences. Since the sequence information retrieved from the GenBank database may not cover all the inter and intra-species DNA sequence variations of the borrelial 16S rRNA genes, the sequence alignments presented in Figure 1 and Figure 2 are to be used as a diagnostic laboratory reference only, and cannot be relied upon for bacterial taxonomy. Multiple gene markers are needed to be considered in taxonomy.

There are more than 40 species in the genus *Borrelia*. Supplementary sequencing of a heminested PCR amplicon (Figure 2) would be able to further speciate many borreliae detected by M1/M2 primer sequencing. An additional sequencing of the 282 bp amplicon can distinguish the less common *B. americana* and *B. californiensis*, but not *B. bissetti*, from *B. burgdorferi* based on alignment of the reference sequences retrieved from the GenBank database; nor would it be able to discriminate between *Borrelia venezuelensis* and *Borrelia* sp. Qtaro, or to distinguish these two borrelial species from *B. turicatae*. However, the lack of ability to speciate the entire *Borrelia* genus should not reduce the usefulness of these “core genome” PCR primers in metagenomic diagnosis of spirochetemia among patients with borreliosis. For the purpose of timely patient management, exact genotyping of all borreliae detected in blood specimens is not necessary. For example, according to established medical practice patients with *Salmonella* septicemia are routinely treated immediately after obtaining a positive blood culture of the pathogen without waiting for the exact final serotyping of the *Salmonella* isolate. For accurate molecular diagnosis of any borrelial isolate, a second gene target may be desirable in addition to 16S rRNA gene sequencing, if possible.

Bacterial 16S rRNA gene is conserved in sequence. Quantitative multiplex real-time PCR (qPCR) of a short segment of borrelial rRNA gene was used to screen the presence of *B. burgdorferi* and *B. miyamotoi* in ticks [51]. In the latter protocol, only random samples scored as *B. miyamotoi* or *B. burgdorferi* by qPCR were confirmed by direct sequencing of the 16S–23S intergenic spacer region (IGR) with species-specific primers. It would be difficult to adapt such a protocol in a clinical laboratory for the diagnosis of all species capable of causing borreliosis; the probe used for qPCR screen would miss *B. valaisiana* and *Borrelia lusitaniae* of the *B. burgdorferi* sensu lato complex due to nucleotide mismatches between the probe and the target sequences in the 16S rRNA gene of these two species. It is noteworthy that the authors of the latter article chose a segment of 16S rRNA gene for screening, but performed direct sequencing of the 16S-23S IGR on only random samples for confirmation [51]. However, the patterns of PCR-restriction fragment length polymorphism (RFLP) of the 16S-23S IGR of *B. burgdorferi* isolates from skin biopsy samples or blood of early Lyme disease patients by cultivation have been found to differ from those assessed by PCR performed directly on patient tissue [52]. The latter inconsistent finding raises the question if 16S-23S IGR can be relied upon as a molecular target for the purpose of direct detection of borrelia in clinical specimens.

Diagnostic real-time PCR is known to be associated with high frequency of false-positives [53]. Since its performance depends on a high ratio of template/non-template DNA in the reaction mixture [54], real-time PCR is not suitable for diagnosis of spirochetemia in Lyme disease patients. The density of borrelia in the circulating blood is usually too low for accurate qPCR diagnostics. *FlaB* gene qPCR was useful in evaluating blood culture of plasma in combination with skin culture [55], but its test results on plasma samples from Lyme disease patients did not significantly correlate with any of the clinical, demographic, or laboratory variables assessed [56]. Until the specificity of qPCR techniques is determined, the clinical utility of such testing relative to other testing modalities will remain uncertain [57].

Metagenomic 16S rRNA gene sequencing is not commonly used for the diagnosis of Lyme and related borreliosis. Free 16S rRNA gene DNA released from lysed bacterial cells is known to be less stable than other genomic DNAs of the bacteria when the source of the DNA is not from a pure culture [58,59]; the mechanism of selective 16S rRNA gene DNA degradation under certain conditions is not clear. We always perform the 16S rRNA gene primary PCR without delay when the DNA extractions from the specimens are available.

Broad-range PCR amplification of a 1343-bp segment and a 762/598 bp segment of bacterial 16S rRNA gene has been used as a tool for DNA sequencing identification of bacteria in pleural fluid and pus in which a plentiful supply of bacterial genomic DNA is available [60]. However, due to low density of borrelia in the circulating blood of the Lyme disease patients with spirochetemia [44,45], there may be only one single copy of target 16S rRNA gene in a PCR mixture as the template for amplification. For the detection of borrelia in blood specimens, the size of PCR amplicons is usually limited to <300 bp to prevent loss of sensitivity [61]. A second PCR to re-amplify the first 357/358 bp primary PCR products with the same M1/M2 PCR primer pair was used for detection in our protocol. However, when the number of PCR runs is increased to a total of 60 thermal cycles, some non-target DNA may be amplified in the absence of borrelial DNA, as shown in Figure 3 and Appendix A and as reported previously [46]. Host DNA interference with PCR detection of *B. burgdorferi* is a well-known phenomenon at least in part because a 21-base long PCR primer is likely to bear more than 50% identity with some of human genomic DNA [62]. In molecular diagnosis, mismatched primers are known to initiate PCR on both target and non-target templates with various degrees of amplification efficiency [63]. It takes 6 matched nucleotides at the 3′ end of a primer with the template to initiate a polymerase chain reaction [64]. As shown in Appendix A, the 21-base *Borrelia* genus-specific M1 primer bears 11 matched nucleotides with a DNA segment on human chromosome 20, including two matched nucleotides at its 3′ terminus (NYS-17 sequence) and bears 9 fully matched nucleotides with a DNA segment on human chromosome 8, all at its 3′ terminus (NYS-18 sequence). In the absence of fully matched borrelia DNA as the preferred template, the M1 and M2 primers may anneal to numerous partially matched human chromosomal DNA fragments to initiate a multi-template PCR [65]. Simultaneous amplification of multiple templates by one pair of PCR primers with multiple partially matched primer-binding sequences on the templates may generate one dominant amplicon in the PCR products to serve as a sequencing template as illustrated in the NYS-17 and NYS-18 electropherograms in Appendix A, or may generate multiple amplicons of slightly different sizes, but close to the size of the target DNA amplicon. In the latter situation, the PCR products may migrate as a band in gel electrophoresis at a size close to that of the target DNA amplicon, but cannot serve as a sequencing template, as illustrated in the NYS-14 sequencing (Appendix A). False-positive PCR amplicons are expected when whole blood samples are used as the starting materials for PCR-based direct detection testing for Lyme disease. Some human nucleated blood cells are often included in the sample prepared for DNA extraction. Therefore, DNA sequencing is mandatory to confirm any visualized PCR amplicon for validation of a molecular diagnosis of spirochetemia.

Detection of residual or leftover free borrelial DNA in circulating blood is not a concern because soluble 16S rRNA gene DNA, if any, in the blood specimens would have been eliminated during differential centrifugation. Our protocol cannot determine if the borreliae detected are live spirochetes, dead spirochetes or only fragments of spirochetes with 16S rRNA gene DNA which are spun down with the platelets at ~16,000× *g*. However, even the presence of fragments of dead borreliae circulating in a patient’s blood is evidence of a recent or an ongoing active infection.

Since a high-fidelity and highly processive DNA polymerase [66] is used in the PCR amplification protocol, pre-PCR sample purification can be eliminated. Using crude NH_4_OH extract to initiate primary PCR reduces the cost of performing PCR tests and avoids the risk of losing the limited copies of target DNA in the sample during purification process. The PCR products generated by the high-fidelity DNA polymerase can be used directly as the template for Sanger reaction without purification, as illustrated in the electropherograms presented in the Appendix A.

### 4.3. Participation of Hospital Laboratories in Endemic Areas for Timely Diagnosis of Spirochetemia

To reduce the burden of borreliosis to society, clinical “Lyme disease” should be diagnosed at the symptomatic spirochetemic stage of early infection, as for any bacteremia and septicemia, at the local hospital laboratories so that the patients can be treated in a timely and appropriate manner in the era of precision medicine. Patients living in an endemic area with suspected clinical manifestations of early “Lyme disease”, such as unexplained sudden headache, low-grade fever, chills, muscle aches, and lymphadenopathy, with or without a skin rash, should be tested for spirochetemia at a local hospital laboratory, rather than having the blood sample sent to geographically distant commercial laboratories for a possible diagnosis of bacteremia. In the past 40 years, Lyme disease has not been seriously studied as an emerging infectious disease among patients as for other newly emerging infectious diseases, such as Ebola [67] and Zika [68] which were diagnosed by nucleic acid amplification tests from the start [67,68]. The laboratory criteria of Ebola virus disease case definition for reporting in the European Union is “Detection of Ebola virus nucleic acid in a clinical specimen and confirmation by sequencing or a second assay on different genomic targets; or isolation of Ebola virus from a clinical specimen” [69]. In contrast, the official CDC Laboratory Criteria for Diagnosis for Lyme disease is “a positive culture for *B. burgdorferi*, or a positive two-tier test, or a positive single-tier IgG WB test for Lyme disease” [70]. However, the CDC also recognizes that serologic assay is unreliable in the diagnosis of bacterial infections, such as using the Widal test for the diagnosis of typhoid fever and states “Serologic assays are not an adequate substitute for blood, stool, or bone marrow culture” [71]. Relying on a single antibody assay to diagnose a complex bacterial infection like borreliosis is bound to generate many questionable and potentially false-negative and false-positive results, leading to many undiagnosed, belatedly diagnosed and misdiagnosed cases of “Lyme disease”. Using the debatable term of “chronic Lyme disease” to characterize the patients with persistent infection due to belatedly diagnosed or undiagnosed borreliosis has been labeled as a “scam” by some infectious disease experts who are skeptical of many existent laboratory tests for Lyme disease [72]. The need for a reliable, irrefutable routine direct detection test based on genomic sequencing is finally recognized by stakeholders of regulatory agencies and the healthcare industry with interests in Lyme disease diagnostics; however, no implementable solution is recommended [22].

Genomic sequencing tests for the diagnosis of “clinical Lyme disease”, or borreliosis as a group, at the early stage of infection have not been explored due to lack of financial supports and lack of implementable diagnostic technologies at the community hospital level where the patients with acute infection are first seen and managed. Our study has presented evidence that one single pair of PCR primers may generate an amplicon to be used as the template for Sanger sequencing for metagenomic DNA detection of the causative agents of borreliosis at the stage of spirochetemia in different endemic areas of the world. Due to sparsity of pathogens in the blood specimens from the Lyme disease patients, many diagnostic laboratories found it challenging in isolating the spirochetes for DNA extraction to initiate a nucleic acid test. We found that a simple differential centrifugation can help overcome this technical difficulty although 100% sensitivity is not achievable in diagnosis of bacteremia with a single test.

Differential centrifugation has been used to concentrate the spirochetes from the platelet rich plasma for DNA extraction while excluding the PCR-inhibiting hemoglobin and human genomic DNA of the whole blood [43,73]. However, it is still not clear what may happen to the limited number of spirochetes in an EDTA whole blood specimen after the blood is drawn from a vein of the patient before the specimen is actually tested in a laboratory. At least one in vitro study has shown that *B. burgdorferi* may actively attach to and invade human lymphocytes during co-incubation of the spirochetes and the blood cells under experimental conditions [74]. A method for optimum sample preparation of the venous blood specimens for metagenomic diagnosis of spirochetemia is still unknown and needs further exploration by the hospital laboratories located in endemic areas dealing with real patients suffering from Lyme disease at different stages of infection. In one summer a collaborative study between the hospital emergency physicians and the laboratory staff in a small hospital located in Connecticut, USA, a Lyme disease-endemic area, where all residents are highly concerned about “catching Lyme disease” in the summer, 5.4% of the patients walking into the emergency room with symptoms suggestive of “Lyme disease” were found to be positive for *B. burgdorferi* spirochetemia by 16S rRNA gene sequencing when the blood specimens were processed within a few hours of venipuncture [75]. In one cold winter month of February, blood specimens from 14 ambulatory patients with a clinical diagnosis of long-term persistent Lyme disease (about 25% of the samples tested) were found to be positive for spirochetemia with 25–50 borrelial cells of either *B. burgdorferi* sensu lato or *B. miyamotoi* per 1 mL of plasma after the spirochetes were spun down in the platelet pellet to be tested [43]. Based on one study of a series of hospitalized patients with acute infection by *B. miyamotoi* or *B. burgdorferi* diagnosed by 16S rRNA gene PCR, the window of opportunity for detecting these spirochetes in the circulating blood seems to be in the first 10 days after onset of symptoms [76]. In the latter study differential centrifugation of the whole blood samples was also used to obtain the platelet-rich plasma for borrelial DNA extraction. The latter model of field study should be replicated in Lyme disease endemic areas worldwide to gain experience in how to properly process blood specimens to increase the detection sensitivity for the diagnosis of spirochetemia.

## 5. Conclusions

In the era of precision medicine, Lyme and related borreliosis, an important but still poorly understood emerging systemic bacterial infection, should be diagnosed by an objective evidence-based laboratory test at the early stage of infection for timely and appropriate patient management to prevent the disease from advancing into a stage of prolonged infection which may lead to serious deep tissue damage. Since the causative agents of borreliosis include various members of the *B. burgdorferi* sensu lato complex and the highly heterogeneous relapsing fever borreliae, a *Borrelia* genus-specific PCR primer pair is needed and has been introduced to amplify a segment of the borrelial 16S rRNA core genome to be used as the template for Sanger sequencing-based metagenomic diagnosis. Examples of computer-generated base-calling sequencing electropherograms used for molecular diagnosis of various borreliae in patient blood specimens and ticks, the arthropod vector of this infectious disease, have been presented to show the feasibility of implementing this diagnostic protocol in clinical laboratories. To be effective in reducing the burden of Lyme disease to society, hospital laboratories located in Lyme disease endemic areas must be actively involved in continued refinement of the methods of diagnosing spirochetemia caused by various species of borreliae.

## Figures and Tables

**Figure 1 ijerph-16-01779-f001:**
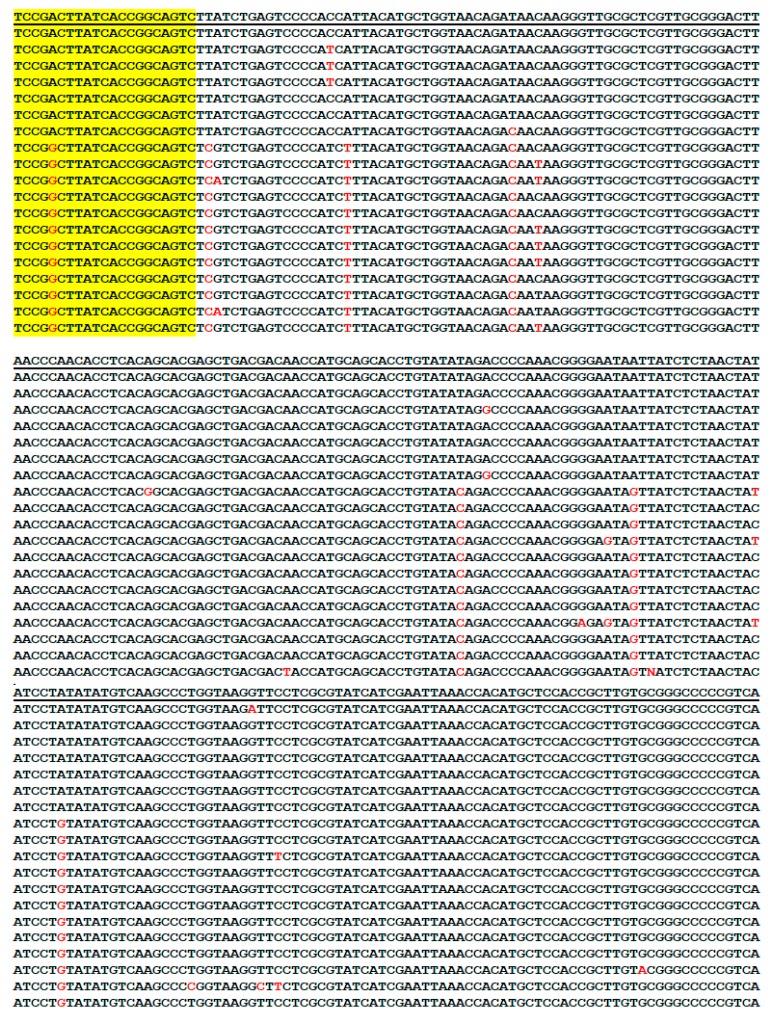
Alignment of 20 highly conserved 357/358-base borrelial 16S rRNA gene segments defined by the M2 and M1 primer sites (yellow-highlighted) with variable regions (typed in red letters). Sequences retrieved from the GenBank database with sequence ID# listed at the end after the M1 primer site. Note: *B.* = *Borrelia*.

**Figure 2 ijerph-16-01779-f002:**
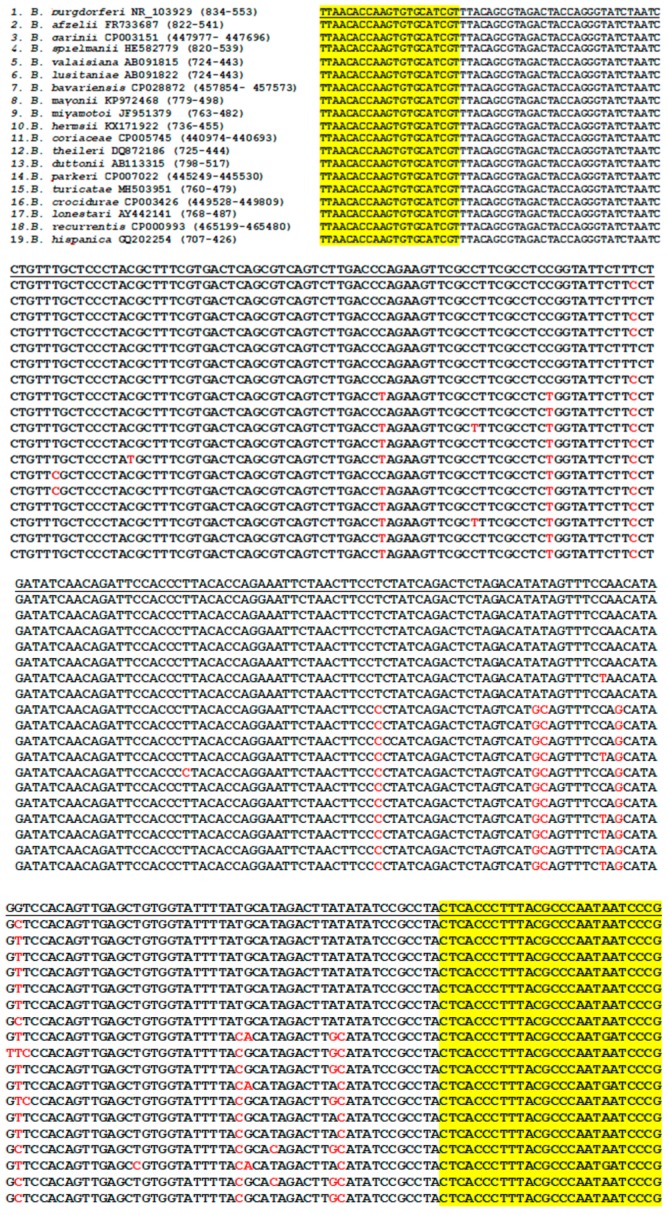
Alignment of the 282-base 16S rRNA gene DNA sequences defined by the Bg6 and Bg5 PCR primer sites (yellow-highlighted) with single nucleotide polymorphisms (in red) of various borrelial species. Sequences with ID# retrieved from the GenBank database listed at the beginning before the Bg6 primer site. The Bg6 primer is the reverse complement of the M1 primer, and this segment of borrelial 16S rRNA gene is an upstream extension of the M1/M2 segment. Note: *B.* = *Borrelia*.

**Figure 3 ijerph-16-01779-f003:**
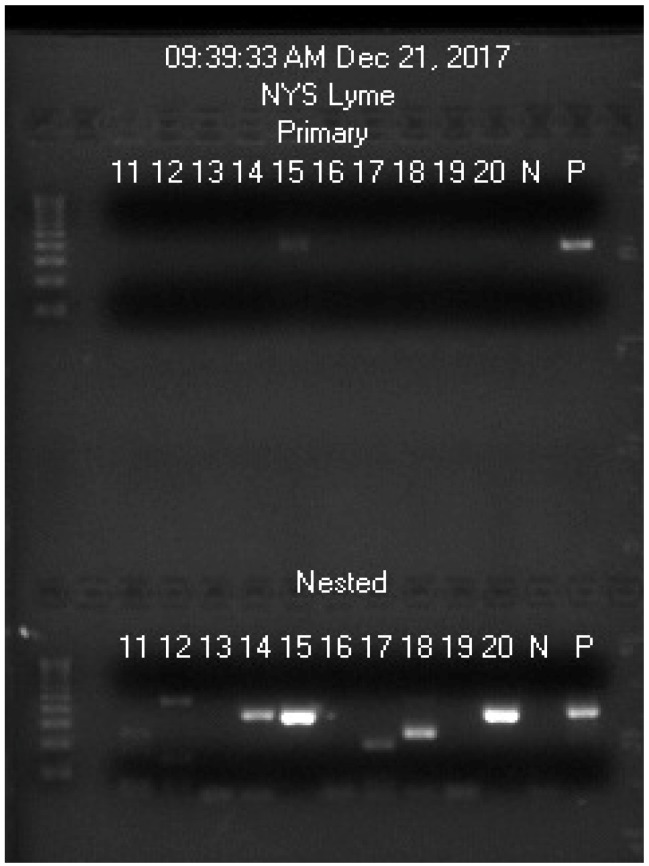
Image of an agarose gel electrophoresis showing 10 primary (upper) and the corresponding second (same-nested) PCR products (lower, labeled as Nested) generated with the M1/M2 primer pair on a panel of blind-coded blood samples received from New York State Department of Health for a *Borrelia burgdorferi* direct detection proficiency test.

**Table 1 ijerph-16-01779-t001:** Decoding of the blinded samples by NYS DOH.

Blood Sample Code No.	11	12	13	14	15	16	17	18	19	20
Spiked with Bb culture	N	N	N	N	Bb	N	N	N	N	Bb

Note: Bb = whole blood spiked with *B. burgdorferi* culture; N = whole blood without borrelia.

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
