# Peer review of "Single Core Genome Sequencing for Detection of both Borrelia burgdorferi Sensu Lato and Relapsing Fever Borrelia Species"

_ijerph, 2019, doi:10.3390/ijerph16101779_

Round 1

Reviewer 1 Report

Review of “Comprehensive Borrelia detection by core genome Sanger sequencing”

Summary of Aim of Paper and Main Contributions

This paper reports on the identification of a highly conserved DNA sequence of the 16S rRNA gene that represents a “core genome” for the Borrelia genus, inclusive of both Bbsl and relapsing fever genospecies. Primers derived from the identified sequences amplified DNA that could be sequenced by the Sanger method for metagenomics analysis. The approach was used to successfully identify Bbsl and relapsing fever borreliae from cultures, from human blood, and from ticks collected from bitten humans and from the environment.  This research identifies a process of sample concentration preceding DNA extraction and amplification that can be applied by clinical or hospital laboratories, to detect patients with borreliosis during the early stages of infection, when standard antibiotic treatment is more likely to be successful.

Strengths

The method described is based on technology (Sanger sequencing) that does not require equipment might be cost prohibitive to small hospitals or clinical laboratories. The paper identifies a “core genome” that is ideal for metagenomics analysis of all pathogenic borreliae in both the Bbsl and relapsing fever subgroups. This is particularly important because the endemicity and human disease risk of the recently identified relapsing fever borreliae B. miyamatoi, which is transmitted by hard-bodies ticks as is Bbsl, remains largely unknown. The described method negates the current issues with the narrow specificity and poor clinical accuracy of the serological tests that are routine used for Lyme disease diagnosis early in the course of the infection by borreliae. Because molecular diagnosis is already being used in diagnostic laboratories, this method could be implemented quickly. The method used for concentration and DNA extraction represents a significant improvement over existing PCR tests for direct detection of borreliosis. DNA sequencing data is included which leaves little room for the findings to be dismissed as “contamination.” The authors review and discuss the “potential pitfalls” of DNA-based diagnostic testing and offer suggestions on how to avoid the problems known to plague this technology in the past. 

Weaknesses

In the Materials and Method section, there is no specific mention of the whether the blood samples used in the study were obtained from people who were informed that it was for a research study. Meaning, did the physicians who submitted the samples have their patients’ sign a form giving their “informed consent” for their blood to be used for research? Presuming that patients were informed their blood was going to be used in this study, how this was done needs to be explicitly stated, in accordance with generally accepted ethical research standards. One sentence should be added to 2.2 Sources of borrelia genomic DNAs stating that the venous blood collected with their informed consent from US patients ….

The primary weakness is compositional and not scientific. The Abstract and Introduction in particular are not concisely written and therefore readers have to search for the pertinent information. The determiner “the” is used far too often and the sentences are way too long, which obscures the nature of the actual problem under investigation. Some specific suggestions to improve the narrative are provided below.

Specific Comments

Lines 11-17: change “before the advent of” to “before nucleic-acid based detection technologies were widely available.” Then start a new sentence – a suggested addition would be to include the US HHS TBDWG reports because it strengthens and legitimized your assertion that serological tests are inaccurate.

Here’s a specific suggestion: “The most widely used diagnostic tests for Lyme disease are based on the serologic detection of antibodies produced against antigens derived from a single strain of Borrelia burgdorferi, only. The poor clinical accuracy of serological tests early in the infection process has been noted, most recently in the 2018 Report to Congress issued by the US Department of Health and Human Services Tick-Borne Disease Working Group (https://www.hhs.gov/ash/advisory-committees/tickbornedisease/reports/index.html). Clinical Lyme disease may be caused by a diversity of borreliae, including those classified as relapsing fever species, in the US and in Europe. It is widely accepted that antibiotic treatment of Lyme disease is most successful during this critical early stage. “

Lines 20-22 – suggest the follow: “…development of a molecular diagnostic tool for all clinical forms of borreliosis have been challenging because a “core genome” shared by all pathogenic borreliae has not yet been identified.”

In general, in the Abstract – remove most uses of “the” as in “…the highly conserved…”  “the common pathogenic borrelia…”  Also simplify “human blood specimens” to “human blood” or just “blood.”

Same comments apply to the Introduction – shorten sentences, use fewer “the’s” – don’t make the reader work to find the relevance of the study and don’t minimize the important concepts (excessive use of passive voice does this).

Line 130-131 – include how patients were advised that their blood was going to be used for research and how they gave their informed consent to the health care providers who submitted their blood for the study.

Author Response

Dear Reviewer 1:

Please see attached Word file response.

Thank you, 

Sin Hang Lee

Reviewer 2 Report

The paper is well written and provides insights in Borrelial molecular diagnostics. It will be further interesting if, the authors could find out some restriction enzyme sites that could be used on the amplified products. However, I understand one can only find what is available by nature.

Author Response

Dear Reviewer 2:

Thank you for your review and comments.. 

Right now, we are focusing on DNA sequencing-based diagnostics because the technology is mature for application, not only for Lyme disease, but also for other infectious diseases. 

Sincerely,

Sin Hang Lee and on behalf of co-authors

Reviewer 3 Report

The study is aimed to establish a Sanger sequencing-based method for clinical diagnosis of Borrelia spp. Two conserved segments (M1-M2 and Bg5-Bg6) were identified in Borrelia 16S rRNA gene as core genomes by alignment of published sequences. In the method M1-M2 or M1-M2 plus Bg5-Bg6 is amplified and followed by Sanger sequencing to detect and differentiate Borrelia. The study evaluated the method with bacterial cultures, clinical samples, and tick samples.

The manuscript is well written with sufficient details in the method and result sections followed by a comprehensive discussion. 

I have the following concerns to the study.

1. The M2 primer has 1 nt mismatch (A->G) with the 9th -20th sequences in Figure 1. Should the M2 primer be designed to a degenerate primer having A or G at that position?

2. 20 species of M1-M2 fragment are aligned in Figure 1 and 19 species (including some species in Figure 2) of Bg5-Bg6 are aligned in Figure 13. Are these all the species that can be amplified by M1/M2 and Bg5/Bg6? Can other species be amplified since Line 549 states “There are more than 40 species in the genus Borrelia”?

3. Have you evaluated coverage of the primers within each species? Sequence variation among strains also should be considered.

4. What criteria were used to determine/assign species? Was it done based on sequence comparison? For example, to assign an isolate into a certain species, the query sequence has to be 100% or >99% identical to the reference sequence. In the Result sections, it seems the critical bases were used to differentiate species. In that case, how conserved the critical bases are among strains in each species?

For example, when a blast search was performed using M1-M2 of B. burgdorferi NR_103929 (the model sequence of B. burgdorferi in Figure 1), there are some strains’ sequences not 100% identical to the fragment of B. burgdorferi NR_103929 such as Genbank ID: X98228.1, X85195.1, X85192.1, and X85189.1.

X98228.1: G at 331 while the base is T in NR_103929

X85195.1: A at 259 while the base is G in NR_103929

X85192.1: C at 37 while the base is T in NR_103929

X85189.1: A at 12 while the base is C in NR_103929

So, when a sequence in above four is obtained and compared with the model sequence of NR_103929, how to determine if it is a B. burgdorferi?

Another example: M1-M2 fragment of many B. garinii strains are 100% identical with the fragment of B. burgdorferi NR_103929. So, Bg5-Bg6 fragment is used as a supplement to further differentiate B. garinii and B. burgdorferi. It is true that almost all B. garinii is T at 207 in the Bg5-Bg6 fragment while B. burgdorferi NR_103929 and many other B. burgdorferi strain are G. Section 3.8 states that G/T difference can be used to distinguish B. garinii and B. burgdorferi. However, many other B. burgdorferi strains are also T at this position such as CP005925.1, CP031412.1, CP017201.1, and CP001205.1. So, the four strains could be misidentified as B. garinii using the G/T differentiation standard.

M1: 5’-ACG ATG CAC ACT TGG TGT TAA-3’

M2: 5’-TCC GAC TTA TCA CCG GCA GTC-3’

Amplicon of B. burgdorferi NR_103929 with M1-M2: 357 bp

TCCGACTTATCACCGGCAGTCTTATCTGAGTCCCCACCATTACATGCTGGTAACAGATAACAAGGGTTGCGCTCGTTGCGGGACTTAACCCAACACCTCACAGCACGAGCTGACGACAACCATGCAGCACCTGTATATAGACCCCAAACGGGGAATAATTATCTCTAACTATATCCTATATATGTCAAGCCCTGGTAAGGTTCCTCGCGTATCATCGAATTAAACCACATGCTCCACCGCTTGTGCGGGCCCCCGTCAATTCCTTTGAGTTTCACTCTTGCGAGCATACTCCCCAGGCGGCACACTTAACACGTTAGCTTCGGTACTAACTTTTAGTTAACACCAAGTGTGCATCGT

Bg5: 5’-CGG GAT TAT TGG GCG TAA AGG GTG AG-3’

Bg6: 5’-TTA ACA CCA AGT GTG CAT CGT-3’

Amplicon of B. burgdorferi NR_103929 with Bg5-Bg6: 282 bp

TTAACACCAAGTGTGCATCGTTTACAGCGTAGACTACCAGGGTATCTAATCCTGTTTGCTCCCTACGCTTTCGTGACTCAGCGTCAGTCTTGACCCAGAAGTTCGCCTTCGCCTCCGGTATTCTTTCTGATATCAACAGATTCCACCCTTACACCAGAAATTCTAACTTCCTCTATCAGACTCTAGACATATAGTTTCCAACATAGGTCCACAGTTGAGCTGTGGTATTTTATGCATAGACTTATATATCCGCCTACTCACCCTTTACGCCCAATAATCCCG

Author Response

Dear Reviewer 3:

Please see attached response from the authors in Word file.

thank you.

Sin Hang Lee 

Round 2

Reviewer 3 Report

The authors have addressed most the comments.